# ENHANCE LOCAL CONSISTENCY FOR FREE: A MULTI-STEP INERTIAL MOMENTUM APPROACH

## ABSTRACT

Federated learning (FL), as a collaborative distributed training paradigm with several edge computing devices under the coordination of a centralized server, is plagued by inconsistent local stationary points due to the heterogeneity of the local partial participation clients, which precipitates the local client-drifts problems and sparks off the unstable and slow convergence, especially on the aggravated heterogeneous dataset. To address these issues, we propose a novel federated learning algorithm, named FedMIM, which adopts the multi-step inertial momentum on the edge devices and enhances the local consistency for free during the training to improve the robustness of the heterogeneity. Specifically, we incorporate the weighted global gradient estimations as the inertial correction terms to guide both the local iterates and stochastic gradient estimation, which can reckon the global objective optimization on the edges' heterogeneous dataset naturally and maintain the demanding consistent iteration locally. Theoretically, we show that FedMIM achieves the $\mathcal{O}(\frac{1}{\sqrt{SKT}})$ convergence rate with a linear speedup property with respect to the number of selected clients $S$ and proper local interval $K$ in each communication round under the nonconvex setting. Empirically, we conduct comprehensive experiments on various real-world datasets and demonstrate the efficacy of the proposed FedMIM against several state-of-the-art baselines.

## 1 INTRODUCTION

Federated Learning (FL) is an increasingly important distributed learning framework where the distributed data is utilized over a large number of clients, such as mobile phones, wearable devices or network sensors (Kairouz et al., 2021). In the contrast to traditional machine learning paradigms, FL places a centralized server to coordinate the participating clients to train a model, without collecting the client data, thereby achieving a basic level of data privacy and security (Li et al., 2020a). The common pipelines to achieve this goal includes three steps (Bonawitz et al., 2019): i) The server broadcasts the current model to clients at the beginning of each communication iteration; ii) The clients synchronize the local models and update the local model based on their own data; iii) The server averages the latest local models and repeats these procedures until convergence.

Despite the empirical success of the past work, there are still some key challenges for FL: expensive communication, privacy concern and statistical diversity. The first two problems are well fixed in past work(Konečnỳ et al., 2016; Sattler et al., 2019; Hamer et al., 2020; Truex et al., 2019; Xu et al., 2019) although the last one is still the main challenge that need to be deal with. Due to statistical diversity among clients within FL system, client drift (Karimireddy et al., 2020a) leads to slow and unstable convergence within model training. In the case of heterogeneous data, each client's optimum is not well aligned with the global optimum. The conventional FL algorithm does not consider this data heterogeneity problem and simply applies the stochastic gradient descent algorithm to the local update. As a consequence, the final converged solution of clients may differ from the stationary point of the global objective function since the average of client updates move towards the average of clients' optimums rather than the true optimum. As the distribution drift exists over the client's dataset, the model may overfit the local training data by applying empirical risk minimization and it has been reported that the generalization performance on clients' local data may exacerbate when clients have different distributions between training and testing dataset (Liang et al., 2020). In order to overcome these problems, several solutions have been put forward in recent years. Generally,

there are three types of methods: variance reduction based (Karimireddy et al., 2020b), regularization based (Li et al., 2020b; Acar et al., 2021) and momentum based (Xu et al., 2021; Reddi et al., 2020). Although these past works present some effective methods to reduce the client drift and improve the generalization performance, the problem of local inconsistency is not fully considered. In the real experiment setting, the local interval $K$ is finite and the local update could not reach the local optimum. With the iteration running, the final points for local iteration will remain relatively stable and become dynamic equilibrium. The stability of these points determines the effectiveness of algorithms and their position will alter when different algorithms are applied. The variance among these points brings the local inconsistency problem Wang et al. (2021a). However, the analysis of these past works are not comprehensive and experimental verification of the reduced local inconsistency is lacking. In particular, when data heterogeneity among clients raises, the local update may repudiate mutually, that is, the direction of the local gradient could not remain compatible. Thus, the weighted average of local gradient at the aggregation stage is extraordinarily small and the moving global iteration point may stagnate, which leads to low generalization performance. To settle this problem, a federated learning algorithm is required to incorporate historical information of full gradient into client local updates for scaling down the variance between local dynamic equilibrium points. Furthermore, the usage of historical full gradient information to navigate the local update ought to be considered wisely instead of simply applied in the weight of models.

In this paper, we develop a new FL algorithm to enhance local consistency for free, Federated Multi-step Inertial Momentum Algorithm (FedMIM), that mitigates client drift and reduces local inconsistency. From a high-level algorithmic perspective, we bring multi-step inertial momentum to the local update, that is, multi-step momentum is placed in both weight (orange arrow shown in Figure 1) and gradient (yellow arrow shown in Figure 1) to modify the local update. Rather than calculating the momentum updates at the server's side and transmitting them through the down-link, all the clients compute the momentum term before the local iteration, while the historical momentum is kept in the client's storage. FedMIM has two major benefits to undertaking aforementioned deficiencies. Firstly, FedMIM does not acquire the server to broadcast the momentum between rounds, which curtails the communication burden. Secondly, in contrast to previous work that focuses on server side momentum (Karimireddy et al., 2020b) or client side momentum Xu et al. (2021), FedMIM delivers inertial momentum term to introduce global information avoiding the gradient exclusion in local update when there exists large data heterogeneity among the participating clients.

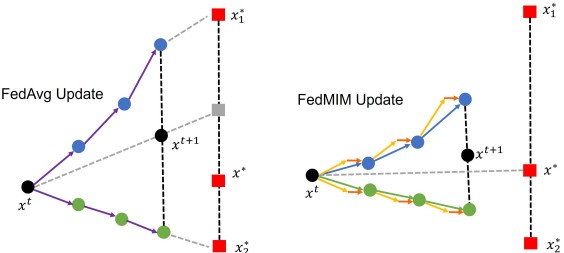

Figure 1: Local steps of FedAvg and FedMIM with 2 clients.

Theoretically, we provide a detailed convergence analysis for FedMIM. By setting proper local learning rate, FedMIM could achieve $\mathcal{O}(\frac{1}{\sqrt{SKT}})$ convergence rate with a linear speedup property for general non-convex setting with the number of selected clients $S$, local interval $K$ and communication round $T$. As for non-convex function under PL condition, convergence rate achieves $\mathcal{O}(\frac{1}{T})$ with proper setting of local learning rate. We test FedMIM algorithm on three datasets (CIFAR-10, CIFAR-100 and TinyImagenet) with i.i.d, and different Dirichlet distributions in the empirical studies. The results display that our proposed FedMIM shows the best performance among the state-of-the-art baselines. When the heterogeneity increases extremely, the performance of the federated algorithms drops rapidly due to the negative impact of enlarging the local interval, while our proposed FedMIM can efficiently maintain stability under the same experimental setups.

**Contribution**. We summarize the main contributions of this work as three-fold:

- FedMIM algorithm delivers a multi-step inertial momentum to guide the gradient updates. We show that FedMIM successfully solves the problems on the heterogeneous datasets, which benefits the cross-device implantation in practical applications.

- We display the convergence analysis of FedMIM for general non-convex function and non-convex function under PL conditions. The theoretical analysis highlights the advantage of innovating multi-step inertial momentum and presents hyperparameter conditions.

- We demonstrate the practical efficacy of the proposed algorithm over competitive baselines through extensive experiments on CIFAR10, CIFAR100 and TinyImagnet datasets, under the various non-i.i.d. dataset splitting, increased local training interval, and different partial participation ratios setups. The results illustrate that FedMIM consistently surpasses several vigorous baselines and significantly handles the local data heterogeneity.

## 2 RELATED WORK

We focus on and categorize the existing approaches to overcoming client drift issues as three main aspects: variance reduction, regularization, and momentum-based algorithms. These methods perform as a global correction to guide the local training, which significantly improves efficiency.

**Variance reduction.** By adopting the variance reduction techniques in the stochastic optimization, a series of methods are proposed to mitigate the heterogeneous inconsistency, which efficiently mitigates the client-drift problems in FL. Karimireddy et al. (2020b) employs control variants to correct for the 'client-drift' in its local updates. Karimireddy et al. (2020a) provides a combination of control-variates and server-level optimizer state at every client-update step to ensure that each local update mimics that of the centralized method. Mitra et al. (2021) includes a gradient correction term in the local update rule that exploits memory. Murata & Suzuki (2021) utilizes small second-order heterogeneity of local objectives and suggests randomly picking up one of the local models instead of taking the average of them when clients are synchronized. Zhao et al. (2021b) applies the vanilla minibatch SGD update or the previous gradient with a small adjustment with pre-defined probability. Zhao et al. (2021a) addresses communication compression and high heterogeneity by proposing compressed and client-variance reduced methods COFIG and FRECON.

**Regularization.** Another efficient approach is to adopt the regularization terms on the local training process to correct the local objective to approach the global optimal. Li et al. (2020b) firstly employs the prox-term into FL framework and propose the FedProx. Tran Dinh et al. (2021) proposes the FedDR with a Douglas-Rachford splitting in the training. Zhang et al. (2020) adopts the primal-dual method to the FL. Acar et al. (2021) improve the FedPD and propose a partial merged parameters method with the full merged dual variables in the global server, named FedDyn. Wang et al. (2022); Gong et al. (2022) adopt the alternating direction method of multipliers in the FL to extend the federated primal-dual methods. Fallah et al. (2020) puts forward a personalized federated framework with the regularization to achieve a better generalization. T Dinh et al. (2020) incorporates the Moreau-Envelopes in the local training with a stage-wised prox-term. Huang et al. (2021) proposes an adaptive weight for the regularization term to encourage the clients to aggregate more with similar neighbours. The efficient regularization methods are important to the FL field.

**Global / Local momentum.** Inspired by the success of the global correction technique, the exponential moving average term is introduced to federated learning framework to correct the local training. Liu et al. (2020) adopts the momentum-SGD to the local clients to improve the generalization performance with a convergence analysis. Wang et al. (2019) proposes a global momentum method to further improve the stability in the server side. Xu et al. (2021) incorporate the global offset to the local client as a client-level momentum to correct the heterogeneous drifts. Ozfatura et al. (2021) combine the global and local momentum update and propose the FedADC algorithm to avoid the local over-fitting. Reddi et al. (2020) sets a global ADAM optimizer with the momentum update and propose the adaptive federated optimizer in the FL. Wang et al. (2021b) corrects the pre-conditioner in the global server. Though momentum terms are the biased estimation of global information, they still contribute a lot to the federated frameworks in practical empirical experiments.

## 3 FEDMIM: FEDERATED MULTI-STEP MOMENTUM ALGORITHM

In this section, we describe how FedMIM works while reducing client drift and improving convergence. To begin with, we provide some preliminary for FL and notations adopted in this paper in Section 3.1. We introduce the diagram of our proposed FedMIM method, and the insights of its improvement on the performance and the resistance to the local heterogeneity in Section 3.2.

### 3.1 PROBLEM SETUP

Considering an FL framework with $N$ local clients and a centralized server to handle the training process. The client $i$ for $i \in [N]$ has the local private dataset $\mathcal{D}_i$ without sharing, and the data sample $\xi_i$ is randomly drawn from the local dataset $\mathcal{D}_i$. The minimization problem could be formulated as:

$$\min_{\mathbf{x} \in \mathbb{R}^d} f(\mathbf{x}) := \frac{1}{N} \sum_{i=1}^{N} f_i(\mathbf{x}) \tag{1}$$

where $f_i(\mathbf{x}) := \mathbb{E}_{\xi_i \sim p_i}[f_i(\mathbf{x}; \xi_i)]$ is the local loss function corresponding to the client $i$ with the data distribution $p_i$. Note that $p_i$ may differ among the local clients, which introduces heterogeneity.

**Notations.** In this paper, we consider $K$ local iteration steps and total $T$ communication rounds in the training. $\|\|$ denotes the Euclidean $l_2$ norm if not otherwise specified. In $t$-th communication round, a set of active clients $\mathcal{S}_t$ with size $S$ is adopted. The symbol $(\cdot)_{i,k}^t$ represents the vectors at $k$-th local step on the $i$-th client after the $t$ communication rounds. For simplicity, $[N]$ represents the set $\{1, 2, \cdots, N\}$. $\mathbf{x}_*$ and $\mathbf{x}_i^*$ stand for global optimum and local optimum for client $i$ respectively.

### 3.2 FEDMIM ALGORITHM

FedAvg proposes a partial scenario of the local SGD method, with the global averaged aggregation after local $K$ updates, disturbed by the client drift problems due to the local heterogeneous case. To mitigate the aforementioned problem, we propose the novel federated learning algorithm with a multi-step inertial momentum-based technique, dubbed Fed-MIM, as shown in the right part. Firstly we introduce the basic process of the proposed FedMIM. At the beginning of each communication round, the server broadcasts the global model $\mathbf{x}_t$ to the activated clients for local training. The local clients update their states $\mathbf{x}_{i,k}^t$ with total $K$ iterations and then transmit their updated models $\mathbf{x}_{i,K}^t$ to the server, while the server aggregates the received local models as the updated global model. In the local clients, they firstly calculate the global model increment $\delta_t$ by the last global model $\mathbf{x}_{t-1}$

---

**Algorithm 1** FedMIM

**Input:** model parameters $\mathbf{x}_0$, constant weight $\{\alpha_j\}_{j \in I}$, constant weight $\{\beta_j\}_{j \in I}$, local learning rate $\eta_l$.
**Output:** model parameters $\mathbf{x}_t$.
1: **for** $t = 0, 1, 2, \cdots, T - 1$ **do**
2:      communicate $\mathbf{x}_t$ to local client $i$ and set $\mathbf{x}_{i,0}^t = \mathbf{x}_t$
3:      randomly select active clients-set $\mathcal{S}_t$ at round $t$
4:      **for** client $i \in \mathcal{S}_t$ parallel **do**
5:          **Local Update:**
6:          $\delta_t = -(\mathbf{x}_t - \mathbf{x}_{t-1})/K$
7:          **for** $k = 0, 1, 2, \cdots, K - 1$ **do**
8:              $\mathbf{y}_{i,k,1}^t = \mathbf{x}_{i,k}^t - \sum_{j \in I} \alpha_j \delta_{t-j} \ (\sum_{j \in I} \alpha_j < 1)$
9:              $\mathbf{y}_{i,k,2}^t = \mathbf{x}_{i,k}^t - \sum_{j \in I} \beta_j \delta_{t-j} \ (\sum_{j \in I} \beta_j = \rho)$
10:            randomly sample local data $\xi_{i,k}^t$
11:            compute stochastic gradient $\mathbf{g}_{i,k}^t$ of $\nabla f_i(\mathbf{y}_{i,k,2}^t)$
12:            $\mathbf{x}_{i,k+1}^t = \mathbf{y}_{i,k,1}^t - (1 - \sum_{j \in I} \alpha_j)\eta_l \mathbf{g}_{i,k}^t$
13:          **end for**
14:          communicate $\mathbf{x}_{i,K}^t$ to the server
15:      **end for**
16:      $\mathbf{x}_{t+1} = \frac{1}{S} \sum_{i \in \mathcal{S}_t} \mathbf{x}_{i,K}^t$
17: **end for**

---

in their local own storage. And then, the clients compute the momentum updated $\mathbf{y}_{i,k,1}^t$ and $\mathbf{y}_{i,k,2}^t$ with a multi-step momentum. It should be noted that $\alpha$ and $\beta$ are adopted for averaging weights in the multi-step momentum term. Each client calculates an unbiased stochastic gradient $\mathbf{g}_{i,k}^t$ and updates its state. When the local update stops, $\mathbf{x}_{i,k}^t$ is transmitted to the server. The iterate scheme details of FedMIM is summarized in Algorithm 1.

**Intuitive Justification** To build intuition into our method, we first highlight multi-step inertial part. Lemma in appendix illustrate that $\delta_t$ is the exponential moving average of past client gradient. The momentum term $\delta_t$ represent as an approximation to the offset of the global loss function $\nabla f(\mathbf{x}_t)$, that is, $\delta_t \approx \eta_l \nabla f(\mathbf{x}_t)$. Thus, we have local update:

$$
\begin{aligned}
\mathbf{x}_{i,k+1}^t &= \mathbf{x}_{i,k}^t - (1 - A)\eta_l \delta_{t-j} - A\eta_l \nabla f_i(\mathbf{x}_{i,k}^t - \eta_l \sum_{j \in I} \beta_j \delta_{t-j}) \\
&\approx \mathbf{x}_{i,k}^t - (1 - A)\eta_l \nabla f(\mathbf{x}_t) - A\eta_l \nabla f_i\left(\mathbf{x}_{i,k}^t - \hat{\rho}\nabla f(\mathbf{x}_t)\right) \\
&\approx \mathbf{x}_{i,k}^t - \eta_l[\nabla f_i(\mathbf{x}_{i,k}^t - \hat{\rho}\nabla f(\mathbf{x}_t)) + (1 - A)(\nabla f_i(\mathbf{x}_{i,k}^t - \hat{\rho}\nabla f(\mathbf{x}_t)) - \nabla f(\mathbf{x}_t))].
\end{aligned} \tag{2}
$$

For simplicity, we set the constant $A = 1 - \sum_{j \in I} \alpha_j$ and $\hat{\rho} = \eta_l \rho$. This equation illustrates that FedMIM interprets the correction term to the local gradient direction. This correction term matches

the difference between global and local gradient. The second term $\nabla f_i(\mathbf{x}_{i,k}^t - \hat{\rho}\nabla f(\mathbf{x}_t))$ in Eq.(2) behaves like Nesterov gradient part, which means that there is global momentum $\hat{\rho}\nabla f(\mathbf{x}_t)$ placed on local iteration point $\mathbf{x}_{i,k}^t$ when client $i$ computes the local gradient in $k$-th local update and $t$-th communication round. The added global momentum pushes the local gradient calculation point to move in the same direction compared with $\nabla f_i(\mathbf{x}_{i,k}^t)$. This benefits the circumstance where the data distribution among the clients differs intensively since the client's update would radiate in a high data heterogeneity environment. In the meanwhile, the added global momentum dwindles gradually as the full gradient $\nabla f(\mathbf{x})$ reduces, and the influence of global momentum scales down with the training process. Therefore, each participating client could reach their dynamic equilibrium at the end of the training. The final correction term is controlled by the parameter $\alpha$. It is notable that the local gradient part is $\nabla f_i(\mathbf{x}_{i,k}^t - \hat{\rho}\nabla f(\mathbf{x}_t))$ rather than $\nabla f_i(\mathbf{x}_{i,k}^t)$ as the direction of local update is $\nabla f_i(\mathbf{x}_{i,k}^t - \hat{\rho}\nabla f(\mathbf{x}_t))$ in the front term and the correction term ought to be consistent with it.

**Discussion**. FedMIM saves the communication bandwidth, which is a crucial problem in FL study. During the broadcasting stage, the server only needs to send the current global state $\mathbf{x}_t$ to the clients rather than the aggregation gradient information in FedCM. The storage of the client is efficiently utilized since it only needs to store $J$ steps of historical information where $J$ is usually set to be very small in practical scenarios and the client's storage requirement does not increase violently. Next, FedMIM simply calculates the gradient once, while FedSAM computes the gradient twice in one local iteration. Thus, the local calculation process could be condensed and total training time is much reduced. The historical global state is stored in clients' storage. FedMIM brings multi-step inertial momentum, which is robust to high client heterogeneity. Since global gradient information is applied to avert the average of client update direction to be minuscule and force global iteration point to move. The introduced multi-step inertial momentum makes the gradient changes more smooth during the local training, although there are some atrocious clients who hold discordant data. The long-step looking makes the approximation exact and smooth for local training, which promotes communication efficiency and enhances the robustness to the heterogeneity in the FedMIM.

## 4 CONVERGENCE ANALYSIS

In this section, we provide the theoretical analysis for FedMIM focusing on the general non-convex setting. Before proposing our convergence analysis, We first state the several assumptions as follows.

**Assumption 1** *For all $\mathbf{x}, \mathbf{y} \in \mathbb{R}^d$, the non-convex $f_i$ is a L-smooth function for all $i \in [N]$, i.e.,*

$$\|\nabla f_i(\mathbf{x}) - \nabla f_i(\mathbf{y})\| \leq L\|\mathbf{x} - \mathbf{y}\| \tag{3}$$

**Assumption 2** *Let $f_* = f(\mathbf{x}_*)$ and $\mathbf{x}_*$ is a minimizer of $f$, for all $\mathbf{x} \in \mathbb{R}^d$, the function $f$ satisfies PL inequality if there exists the constant $\mu > 0$ such that the function $f$ satisfies the following:*

$$\frac{1}{2}\|\nabla f(\mathbf{x})\|^2 \leq \mu(f(\mathbf{x}) - f_*) \tag{4}$$

**Assumption 3** *For all $\mathbf{x} \in \mathbb{R}^d$, the stochastic gradient $\nabla f_i(\mathbf{x}, \xi)$, computed by the sampled data $\xi$ on the local client $i$, is an unbiased estimator of $\nabla f_i(\mathbf{x})$ with bounded variance $\sigma_l^2$, i.e.,*

$$\mathbb{E}_\xi[\nabla f_i(\mathbf{x}, \xi)] = \nabla f_i(\mathbf{x}), \quad \mathbb{E}_\xi\|\nabla f_i(\mathbf{x}, \xi) - \nabla f_i(\mathbf{x})\|^2 \leq \sigma_l^2 \tag{5}$$

**Assumption 4** *For all $\mathbf{x} \in \mathbb{R}^d$, the local functions $f_i$ holds $(G, B)$-locally dissimilarity with $f$, i.e.,*

$$\frac{1}{N}\sum_{i=1}^N \|\nabla f_i(\mathbf{x})\|^2 \leq G^2 + B^2\|\nabla f(\mathbf{x})\|^2. \tag{6}$$

These assumptions are commonly used in federated optimization (Li et al., 2020b; Reddi et al., 2020; Karimireddy et al., 2020a;b). Assumption 1 tells the smoothness of local loss function $f_i$, that is, the gradient function of $f_i$ is Lipschitz continuous with Lipschitz constant $L$. Assumption 2 shows the global function satisfies the PL conditions. The PL inequality does not require $f_i$ to be convex but suggests that every stationary point is a minimum. The $\mu$-PL property is implied by $\mu$-strong convexity, but it allows for multiple minima and does not require convexity of any kind.

Assumption 3 bounds the variance of stochastic gradient and Assumption 4 provides the bound of the different levels of the local private heterogeneity.

We now state our convergence results of FedMIM. The detailed proof is stated in Appendix.

**Theorem 4.1** *Let Assumptions 1, 3, and 4 hold. Assume the partial participation ratio being $|\mathcal{S}_t|/N$ where $\mathcal{S}_t$ is a uniformly sampled subset from the $N$ clients and satisfies $|\mathcal{S}_t| = S$ and let $\mathbf{u}_t = \mathbf{x}_t - \frac{K}{A}\sum_{j \in I}\sum_{s=j}^{J-1}\alpha_s\delta_{-j}$. With the constant local learning rate satisfying $\eta_l \leq \min\{\frac{1}{4LK\sqrt{A}}, \frac{3}{16KL}\}$ and $\lambda \in (0, \frac{1}{2})$ in Algorithm 1, the sequence $\{\mathbf{u}_t\}$ satisfies the following upper bound:*

$$\min_{t \in [T]} \mathbb{E}\|\nabla f(\mathbf{u}_t)\|^2 \leq \frac{f_0 - f_*}{\eta_l \lambda KT} + \Psi$$

*where $\Psi = \frac{1}{\lambda}\left(\frac{\eta_l L\sigma_l^2}{S} + \frac{4\eta_l KLG^2}{S} + 9\eta_l^2 A^2 KL^2\sigma_l^2 + 72\eta_l^2 AK^2 L^2 G^2 + 3\eta_l^2 L^2 K^2 VC\right)$. $V, C$ are two constants defined in the proof for the convergence analysis (details are stated in the Appendix).*

**Remark 4.1** *If the number of total clients $N$ is large enough, the initial state point will affect the convergence upper bound to a great extent, which requires a larger local learning rate $\eta_l$ to diminish the negative impact. Specifically, when we fix $N$ as a constant and select a proper local interval $K$, let $\eta_l = \mathcal{O}(\frac{\sqrt{S}}{\sqrt{KT}})$, the convergence rate achieves at least $\mathcal{O}(\frac{1}{\sqrt{SKT}})$, which indicate the linear speedup of the FedMIM and the stochastic variance dominates the upper bound of the convergence.*

**Theorem 4.2** *Let Assumption 1, 2, 3, and 4 hold and all the conditions being similar as required by Theorem 4.1. Given $\eta_l \geq \frac{1}{\mu\lambda KT}$, the output $\mathbf{u}^{\text{out}}$ chosen randomly from the sequence $\{\mathbf{u}_t\}$ satisfies:*

$$\mathbb{E}\left\|\nabla f\left(\mathbf{u}^{\text{out}}\right)\right\|^2 \leq 4\mu(f_0 - f_*)e^{-\mu\eta_l \lambda KT} + \Psi$$

*where $\Psi = \frac{1}{\lambda}\left(\frac{2\eta_l L\sigma_l^2}{S} + \frac{8\eta_l KLG^2}{S} + 18\eta_l^2 KA^2 L^2\sigma_l^2 + 144\eta_l^2 AK^2 L^2 G^2 + 6\eta_l^2 L^2 K^2 VC\right).$*

**Remark 4.2** *The term introduced by initial point is exponential diminished by the communication round $T$. Let $\eta_l = \mathcal{O}(\frac{\log(\mu^2 ST)}{\mu\lambda KT}) \geq \frac{1}{\mu\lambda KT}$, the convergence rate achieves at least $\mathcal{O}(\frac{1}{\mu ST})$.*

**Remark 4.3** *The $B$ in Assumption 4 weakly influences the convergence bound both in Theorem 4.1 and 4.2 in our proof, which indicates that the major negative impact from the heterogeneity is the constant upper bound $G$. If $G$ maintains the stability without large fluctuations during the training, let $\mathbf{x} = \mathbf{x}_*$ we have $\frac{1}{N}\sum_{i=1}\|\nabla f_i(\mathbf{x}_*)\|^2 \leq G^2$, where $G$ measures the local inconsistency of total clients. Enhancing the local consistency will further improve the performance in the FL framework.*

## 5 EXPERIMENTS

In this section, we demonstrate the efficacy of the proposed FedMIM. We test the generalization performance under different levels of the heterogeneity on the real-world dataset. To ensure a fair comparison, we fix all the common hyper-parameters and finetune the specific parameters unique to each algorithm to search for their best performance. We provide a brief introduction to the experimental setups in 5.1. We compare the proposed FedMIM with the baselines and report their performance in 5.2. Some ablation studies and hyper-parameters sensitivity studies are stated in 5.3.

### 5.1 SETUPS

**Dataset.** We conduct the extensive experiments on CIFAR-10, CIFAR-100 (Krizhevsky et al., 2009) and TinyImagenet (Le & Yang, 2015). Both CIFAR-10 and CIFAR-100 contain 50K training samples and 10K test samples of images with the size of $32 \times 32$. TinyImagenet contains 200 categories of 100K training samples and 10K test samples of images with the size of $64 \times 64$ selected from the Imagenet (Deng et al., 2009). We divide the training dataset into $N$ parts and deploy them to local clients without sharing access. At the beginning of each communication round in the training, we randomly crop, horizontally flip, and normalize the local dataset as the common data augmentation.

**Heterogeneity.** For the IID setting, the local dataset is randomly sampled. For the non-IID setting, we follow by Hsu et al. (2019) to introduce different levels of the heterogeneity by sampling the label ratios from different Dirichlet distributions, which is a common federated setup in previous works (Reddi et al., 2020; Karimireddy et al., 2020b; Acar et al., 2021; Xu et al., 2021; Qu et al., 2022; Kim et al., 2022). In addition, we superimpose a color perturbation (Arjovsky et al., 2019) which is strongly correlated to the local clients to further induce the heterogeneity. Specifically, we adopt different brightness and saturation coefficients to the different local training data samples.

**Baselines.** We compare the performance of several SOTA baselines, including FedAvg (McMahan et al., 2017), FedAdam (Reddi et al., 2020), SCAFFOLD (Karimireddy et al., 2020b), FedDyn (Acar et al., 2021), FedCM (Xu et al., 2021), and FedSAM (Qu et al., 2022), on the backbone of standard ResNet-18 network implemented in the Pytorch Model Zoo ($7 \times 7$ filter in the 1st conv) (He et al., 2016) with the group normalization (Wu & He, 2018; Hsieh et al., 2020). We summarize and discuss these methods in Section 3.2 to illustrate the respective improvements and practical performance.

**Hyper-parameters selections.** To ensure a fair comparison, we fix the common hyper-parameter setups. We set the local learning rate as $0.1$ and decay it as $0.998\times$ per round. The global learning rate is set as $1.0$ to aggregate each local parameters without decaying, except for FedAdam which adopts $0.1$. The local mini-batch is selected in $\{20, 50\}$. The weight decay is set as $1e\text{-}3$. The local training epoch is selected in $\{1, 2, 5, 10\}$ to further show the impacts of enlarging the local interval. The number of total clients is selected in $\{100, 500\}$ and the sampling probability of each client being activated per communication round is selected in $\{0.2, 0.1, 0.02\}$. The prox-weight in FedDyn and the client-level momentum weight in FedCM are both set as $0.1$. We report the detailed hyper-parameters selections for each experimental result in the following figures and tables.

## 5.2 EXPERIMENTAL RESULTS

### 5.2.1 COMPARISON ON INCREASING THE HETEROGENEITY.

To explore the impact of introducing the heterogeneity, we select the three splitting methods on the dataset, including IID, Dirichlet-0.3 and Dirichlet-0.1. On the simple CIFAR-10/100 dataset, FedMIM achieves top-1 performance among the three heterogeneous settings. In the IID case on CIFAR-10, FedMIM achieves $86.39\%$ with $4.23\%$ over the FedAvg baseline. The second top performance of FedCM is $85.62\%$. When the heterogeneity is increased to DIR-0.3, FedCM drops from $85.62\%$ to $82.39\%$ with $3.23\%$ loss, while FedMIM drops only $2\%$ and maintains the top-1 accuracy. The other methods like SCAFFOLD, FedDyn and FedSAM are affected at different levels. When we further enlarge the heterogeneity to DIR-0.1, the FedAdam is most affected and its accuracy drops from $83.19\%$ to $71.75\%$ with an approximate $12\%$ loss. On the large CIFAR-100 and TinyImagenet datasets, similar results can be observed. Our proposed FedMIM have the very stable test accuracy. In particular, when the heterogeneity is introduced to DIR-0.1 on CIFAR-100, FedMIM achieves only $1\%$ drops, which is far better than the others with at least $2\%$. Its performance is also better than the test accuracy on DIR-0.3 of most other baselines. In the IID splitting of TinyImagenet, the FedAdam even can achieve the top-1 performance, while when the heterogeneity its performance drops rapidly and even worse than FedAvg. FedMIM adopts the inertial momentum to the local training both on the iteration points and gradients and enhances the local consistency, which can efficiently resist on the heterogeneity. The multi-step makes the gradient changes more smooth during the training, even under the participation of some bad samples of clients whose dataset holds a very large difference, the long-step looking makes the approximation exact and stable for local training, which encourages the efficiency and robustness to the heterogeneity for the FedMIM.

### 5.2.2 COMPARISON ON ENLARGING THE LOCAL INTERVAL.

To further explore the impact of enlarging the local interval $K$, we select the three different local intervals to test the performance of our proposed FedMIM and the other baselines. We follow the previous works (Acar et al., 2021; Xu et al., 2021; Kim et al., 2022) to compare the performance under different local epochs $E = 1, 5, 10$. The total training samples in CIFAR-10/100 is 50,000 and they are split into 100 parts equally with 500 samples from a local client. To fairly compare with the others, we fix the batchsize as 50, which means the local iteration is TrainSamples/Batchsize $= 10$ per epoch. It should be noted that in the proof the local interval $K$ corresponds to the iteration. When the $E = 1$ with a short local interval, local training do not introduce more local heterogeneity

Table 1: Test accuracy (%) after 1000 communication rounds on IID., Dirichlet-0.3 (DIR.3), and Dirichlet-0.1 (DIR.1) dataset. We set the number of clients as 100 and set the participation ratio as 0.1. The local interval is fixed as 5 epochs and the batchsize is fixed as 50 for all algorithms.

| Method | CIFAR-10 | | | CIFAR-100 | | | TinyImagenet | | |
|---|---|---|---|---|---|---|---|---|---|
| | IID. | DIR.3 | DIR.1 | IID. | DIR.3 | DIR.1 | IID. | DIR.3 | DIR.1 |
| FedAvg | 82.16 | 80.19 | 75.92 | 42.77 | 42.18 | 40.88 | 30.31 | 28.61 | 27.93 |
| FedAdam | 83.19 | 78.61 | 71.75 | 48.92 | 44.58 | 42.96 | **37.04** | 32.34 | 29.44 |
| SCAFFOLD | 84.43 | 82.39 | 78.78 | 49.11 | 48.34 | 46.93 | 35.09 | 34.68 | 34.17 |
| FedCM | 85.62 | 82.57 | 77.31 | 51.14 | 49.27 | 45.59 | 34.35 | 33.34 | 32.34 |
| FedDyn | 84.23 | 81.44 | 75.66 | 46.87 | 45.77 | 43.93 | 33.91 | 31.97 | 30.11 |
| FedSAM | 84.98 | 82.13 | 77.65 | 48.32 | 46.24 | 44.18 | 35.04 | 33.15 | 32.86 |
| **Ours** | **86.39** | **84.39** | **80.82** | **52.83** | **50.53** | **49.20** | 36.47 | **35.17** | **34.83** |

Table 2: Test accuracy (%) after the corresponding proper communication rounds and local epoch $E = 1, 5, 10$ respectively. We set the number of clients as 100 and the participation ratio as 0.1. The batchsize is fixed as 50 and the heterogeneous dataset is divided as the Dirichlet-0.6 distribution.

| Method | CIFAR-10 | | | CIFAR-100 | | | TinyImagenet | | |
|---|---|---|---|---|---|---|---|---|---|
| | E = 1 | E = 5 | E = 10 | E = 1 | E = 5 | E = 10 | E = 1 | E = 5 | E = 10 |
| FedAvg | 82.76 | 81.84 | 81.06 | 44.95 | 42.45 | 40.84 | 36.28 | 28.79 | 24.77 |
| FedAdam | 83.56 | 81.42 | 80.21 | 51.50 | 46.16 | 44.18 | 38.25 | 32.14 | 21.89 |
| SCAFFOLD | 85.03 | 84.12 | 83.54 | 48.35 | 48.85 | 43.25 | 37.21 | 34.87 | 26.75 |
| FedCM | 86.33 | 84.89 | 82.83 | 52.11 | 50.42 | 48.14 | 40.63 | 33.70 | 25.82 |
| FedDyn | 85.45 | 83.96 | 78.12 | 49.94 | 45.80 | 44.97 | 39.33 | 30.55 | 22.36 |
| FedSAM | 84.48 | 84.17 | 82.97 | 49.18 | 46.79 | 45.33 | 38.11 | 32.94 | 25.16 |
| ours | **86.68** | **85.18** | **84.13** | **55.77** | **51.37** | **50.13** | **40.95** | **36.82** | **28.32** |

to the global view. When the local epochs are enlarged to 10, the long local update exacerbates the inconsistency problem and shows a negative impact on the test accuracy. Especially on the large TinyImagenet dataset, most algorithms fail to converge at $T = 1000$. Thus the test accuracy could be considered as the convergence rate for all the methods. FedMIM achieves the top-1 accuracy on both short epochs and long epochs. In the local iteration, the inertial momentum which promotes the local consistency, plays an important role in the stochastic gradient estimation. FedMIM obtains the iterative point closer to the global iterative point via perturbing the local gradient, which approximates the global direction and updates it by one step gradient descent. This allows the local update to be corrected not only on the gradient term, but also on the iterative points where the gradient is calculated. In the next part, we will discuss the consistency between the baselines and some ablation studies, including the participation ratio, the selection of the $\alpha_j$ and $\beta_j$ and the different multi-steps.

## 5.3 ABLATION STUDIES AND HYPER-PARAMETER SENSITIVITY

### 5.3.1 PARTICIPATION RATIO

We test the experiments on the CIFAR-10 dataset under different participation ratios, which are selected from $5\%, 20\%$ to test the convergence rate under the setups of fixed local epoch 5 and batchsize 50. The heterogeneity is set as DIR-0.1. From the Figure 2 (a), when the heterogeneity is enlarged, the convergence speed of FedAvg loses the most performance. FedAdam, FedCM, and FedDyn show a high sensitivity to the participation ratios. SCAFFOLD performs well and maintains the excellent generalization performance via the variance reduction technique under the higher participation option. Beneficial from the inertial momentum, the global direction could be exact estimated. And a multi-steps calculation is adopted to further enhance the stability and smooth characteristic in the estimation. Our proposed FedMIM shows a very stable performance both on different heterogeneity and participation ratios, especially on the extreme heterogeneous settings.

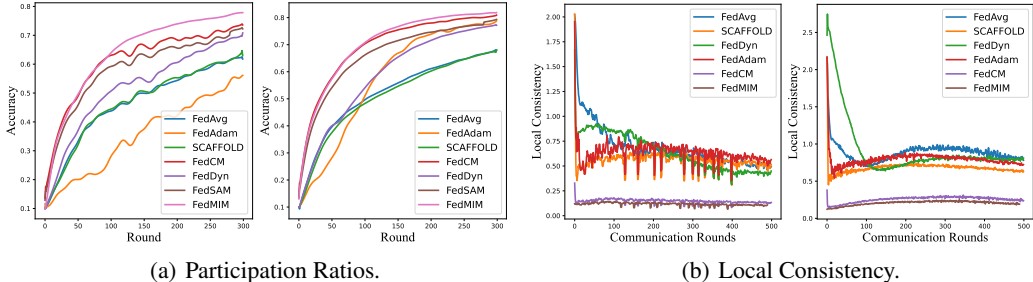

|                     (a) Participation Ratios.                     |                    (b) Local Consistency.                    |

Figure 2: (a) Comparison on different participation ratios on CIFAR-10 dataset and (b) comparison on different methods for the local consistency. We fix the other hyper-parameters. In (a), $L/R$ shows the ratio equals to $5\%/20\%$. In (b), $L/R$ shows the consistency under the CIFAR10/100 dataset.

### 5.3.2 SELECTION OF MULTI-STEPS

We test the different selection of steps $J$ and selection of $\alpha_j$, $\beta_j$. The experimental setup is: local epochs 5, total communication rounds 500 and batchsize is fixed as 50. When $J = 2$, FedMIM achieves the best generalization performance. As shown in Table 3, if we set $\alpha_j = 0$ and $\beta_j = 0$, FedMIM degenerates to the FedAvg method. And if we set $\alpha_2 = 0$ and $\beta_j = 0$, FedMIM degenerates to the FedCM method. It shows that the $\beta_j$ with a long history is not a good selection for the local clients, due to the expired information before the current time. Local updates will be misled by

Table 3: Selection of $\alpha_j$ and $\beta_j$.

| $\alpha_1$ | $\alpha_2$ | $\beta_1$ | $\beta_2$ | Accuracy(%) |
|---|---|---|---|---|
| - | - | - | - | 81.61 |
| 0.9 | - | - | - | 84.14 |
| 0.5 | - | - | - | 83.53 |
| 0.8 | 0.1 | - | - | 84.29 |
| 0.6 | 0.3 | - | - | 84.67 |
| 0.3 | 0.6 | - | - | 83.86 |
| **0.6** | **0.3** | **0.9** | **0.1** | **84.98** |
| 0.6 | 0.3 | 0.5 | 0.5 | 84.42 |
| 0.6 | 0.3 | 0.1 | 0.9 | 84.04 |

the redundancy of the invalid offset. The adjacent update is the most important. While the $\alpha_j$ is more relaxed, which can be searched from the last two or three steps. In the empirical studies, we recommend the selection can be decided by different indicators, a large $\alpha_j$ and a proper $\beta_j$ are better.

### 5.3.3 LOCAL CONSISTENCY

We test the consistency during the training as $\frac{1}{S} \sum_i \|\mathbf{x}_{i,K}^t - \mathbf{x}^{t+1}\|^2$ where $\mathbf{x}^{t+1} = \frac{1}{S} \sum_i \mathbf{x}_{i,K}^t$. In the practical training, the local models can not approach the true local optimal due to the limitation of local interval $K$, thus all the $\mathbf{x}_{i,K}^t$ will represent for the dispersion from the global model $\mathbf{x}^{t+1}$. To keep the $\mathbf{x}_{i,K}^t$ close to each other can improve the resistance to the local heterogeneity (the idealized case is that all local clients always generate the same parameters per round). Figure 2 (b) shows the empirical results of the consistency on the different dataset, FedMIM handles the more excellent efficiency on maintaining the local similarity than the other baselines on the both CIFAR-10/100.

## 6 CONCLUSION

In this work, we propose a novel federated algorithm, named FedMIM, which adopts the multi-step inertial momentum to guide the local training on the heterogeneous clients both on the gradient estimation and the iterative point for gradient calculation. We also theoretically prove that the proposed FedMIM achieves $\mathcal{O}\left(\frac{1}{\sqrt{SKT}}\right)$ under the smoothness assumptions and $\mathcal{O}(\frac{1}{T})$ under the Polyak-Lojasiewicz (PL) inequality, under the non-convex cases. FedMIM can efficiently improve the local consistency to mitigate the influence from the heterogeneous dataset. We conduct extensive experiments to demonstrate the significant performance of our proposed FedMIM on the real-world dataset. Furthermore, we learn some ablation studies to verify the stability under different setups.

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
