# OpenReview forum: "Enhance Local Consistency for Free: A Multi-Step Inertial Momentum Approach"
_ICLR.cc/2023/Conference — Submitted to ICLR 2023_

### Official Review · Reviewer_WgvB · 2022-10-24

**Confidence:** 3
**Correctness:** 4
**Technical Novelty And Significance:** 2
**Empirical Novelty And Significance:** 2
**Recommendation:** 5

**Clarity, Quality, Novelty And Reproducibility:**

lack on novelty

lack of code to reproduce experiment results

**Strength And Weaknesses:**

strength: multi-step inertial momentum on the edge devices to enhances the local consistency

weakness: lack of novelty compared to the server side momentum (Karimireddy et al., 2020b) or client side momentum Xu et al. (2021)

**Summary Of The Paper:**

This work propose a federated learning algorithm, named FedMIM, which adopts the multi-step inertial momentum on the edge devices and enhances the local consistency for free during the training to improve the robustness of the heterogeneity. Specifically, the authors incorporate the weighted global gradient estimations as the inertial correction terms to guide both the local iterates and stochastic gradient estimation, which can reckon the global objective optimization on the edges’ heterogeneous dataset naturally and maintain the demanding consistent iteration locally.

**Summary Of The Review:**

federated learning is a great area to study. this work adopts the multi-step inertial momentum on the edge devices to improve the robustness of the heterogeneity.  However, the novelty of the paper is not well-established.

---

### Official Review · Reviewer_YuXz · 2022-10-25

**Confidence:** 4
**Correctness:** 3
**Technical Novelty And Significance:** 2
**Empirical Novelty And Significance:** 2
**Recommendation:** 3

**Clarity, Quality, Novelty And Reproducibility:**

The writing can significantly be improved.
The novelty of this paper is limited. This paper basically follows the existing Nesterov' momentum techniques.
The motivation of this paper is not clear.


**Strength And Weaknesses:**

Strength:

This paper proposed a new federated learning algorithm (i.e., FedMIM),  which adopts the multi-step inertial momentum on the edge devices and  enhances the local consistency for free during the training  to improve the robustness of the heterogeneity. Theoretically, it proved that
the FedMIM algorithm achieves the best known convergence rate with a linear speedup property with respect to the number of selected clients and proper local interval in each communication round under the nonconvex setting.

Weakness:

The novelty of this paper is limited. This paper basically follows the existing Nesterov' momentum techniques.

**Summary Of The Paper:**

This paper proposed a new federated learning algorithm (i.e., FedMIM),  which adopts the multi-step inertial momentum on the edge devices and enhances the local consistency for free during the training  to improve the robustness of the heterogeneity. Theoretically, it proved that
the FedMIM algorithm achieves the best known convergence rate with a linear speedup property with respect to the number of selected clients and proper local interval in each communication round under the nonconvex setting. Empirically, it conducted comprehensive experiments on various real-world datasets and demonstrate the efficacy of the proposed FedMIM against several state-of-the-art baselines.


**Summary Of The Review:**

This paper proposed a new federated learning algorithm (i.e., FedMIM),  which adopts the multi-step inertial momentum on the edge devices and  enhances the local consistency for free during the training  to improve the robustness of the heterogeneity. Theoretically, it proved that  the FedMIM algorithm achieves the best known convergence rate with a linear speedup property with respect to the number of selected clients and proper local interval in each communication round under the nonconvex setting. Empirically, it conducted
comprehensive experiments on various real-world datasets and demonstrate the efficacy of the proposed FedMIM against several state-of-the-art baselines.

Some Comments:

1. The motivation of this paper is not clear. Why could multi-step inertial momentum on the edge devices
 enhance local consistency ?

2. What is the parameter $I$ in the FedMIM algorithm ？

3. How to choose the tuning parameters $\alpha_j$ and $\beta_j$ in the FedMIM algorithm ?

4. Reproducibility is a crucial aspect of any work based on experiments.
Your code should be available to the reviewers throughout all stages of the submission and publicly.

---

### Official Review · Reviewer_qspk · 2022-10-28

**Confidence:** 4
**Clarity, Quality, Novelty And Reproducibility:** Please see the above review for furth…
**Correctness:** 4
**Technical Novelty And Significance:** 4
**Empirical Novelty And Significance:** 4
**Recommendation:** 6

**Strength And Weaknesses:**

The paper is well written and the main contributions of the work are well presented. I went through the main proofs and the important steps looks also correct.

The paper is easy to follow and the main contributions are clear.

Suggestion: It would be nice at the point where FedMIMN proposed a pseudocode to explain how one can obtain the classical Local SGD as a special case (if this is possible).

Missing references: Closely related papers are [1] and [2] below.

Questions:
1. Also to the best of my knowledge one of the state-of-the-art methods for solving non-convex problems is the $\tau$-overlap SGP from paper [2]. How your approach is related to this method?

2. SlowMo from the paper Wang et al. (2019), which is already cited in the paper, is also a fast method. Why is not used in the experiments?

Missing References:

[1] Koloskova, Anastasia, Nicolas Loizou, Sadra Boreiri, Martin Jaggi, and Sebastian Stich. "A unified theory of decentralized sgd with changing topology and local updates." In International Conference on Machine Learning, pp. 5381-5393. PMLR, 2020.

[2] Assran, Mahmoud, Nicolas Loizou, Nicolas Ballas, and Mike Rabbat. "Stochastic gradient push for distributed deep learning." In International Conference on Machine Learning, pp. 344-353. PMLR, 2019.

**Summary Of The Paper:**

The paper proposes a new federated learning algorithm, named FedMIM, which adopts the multi-step inertial momentum
on the edge devices and enhances the local consistency for free during the training to improve the robustness of the heterogeneity.

In terms of theory the paper, the paper proves that FedMIM achieves the $O(1/\sqrt{SKT})$ convergence rate under the nonconvex setting, where $S$ is the number of selected clients and $K$ is the proper local interval in each communication round. For non-convex function under PL condition, a convergence rate $O(1/T)$ was also proved for FedMIM.

The paper conduct comprehensive experiments on various real-world datasets and demonstrate the efficacy of the proposed FedMIM against several state-of-the-art baselines.

**Summary Of The Review:**

As I mentioned above, the paper is well written and the main contributions of the work are well presented.

I give a score of "6: marginally above the acceptance threshold" to this submission.

---

### Decision · Program_Chairs · 2023-01-20

**Decision:**

Reject

**Justification For Why Not Higher Score:**

Lack of novelty in setting or algorithm, similarity to many previously published works.

**Justification For Why Not Lower Score:**

N/A

**Metareview: Summary, Strengths And Weaknesses:**

This paper proposes a new variant of federated learning with momentum, establishes its convergence rate and shows its empirical performance. The main hinderance to a higher score for this paper is the lack of novelty in either the problem setting, the algorithm or its analysis.

This is reflected both in the comments of the reviewers, and in their scores. There is now a rich enough body of literature with similar algorithms and analyses that with this exact setup it may be hard to stand apart even if it is well written and technically sound.

**Summary Of Ac-Reviewer Meeting:**

N/A